# Parker’s Solar Wind Model for a Polytropic Gas

**DOI:** 10.3390/e23111497

**Published:** 2021-11-12

**Authors:** Bhimsen Shivamoggi, David Rollins, Leos Pohl

**Affiliations:** 1Department of Mathematics, University of Central Florida, Orlando, FL 32816, USA; drollins525@gmail.com; 2Department of Physics, University of Central Florida, Orlando, FL 32816, USA; lpohl@ucf.edu

**Keywords:** Parker solar wind, polytropic flows

## Abstract

Parker’s hydrodynamic isothermal solar wind model is extended to apply for a more realistic polytropic gas flow that can be caused by a variable extended heating of the corona. A compatible theoretical formulation is given and detailed numerical and systematic asymptotic theoretical considerations are presented. The polytropic conditions favor an enhanced conversion of thermal energy in the solar wind into kinetic energy of the outward flow and are hence shown to enhance the acceleration of the solar wind, thus indicating a quicker loss of the solar angular momentum.

## 1. Introduction

Stellar wind is a continuous plasma outflow from a star. In the case of the Sun, this outflow typically emerges from coronal holes (Sakao et al. [1]) and carries a remnant of the stellar magnetic field that fills the space around the star (in the case of the Sun, this constitutes the heliosphere (Dialynas et al. [2])). (Note: SOHO observations (Cho et al. [3]) revealed that the structure of the solar wind changes from the solar maximum to the solar minimum period.) Stellar winds carry off, especially when magnetized, a huge amount of angular momentum from the stars while causing a very negligible amount of mass loss from the stars. Weak to moderate stellar winds are generated by an expanding outer corona due to an extended active heating of the corona in conjunction with high thermal conduction. (Note: Parker [4,5,6] proposed that this is caused by the dissipation of plasma waves produced by micro-flares in the coronal holes. The details of the coronal heating mechanism are still controversial (Klimchuk [7]).) However, control of coronal base conditions and high thermal conduction are both inadequate to generate observed high wind speeds in the case of the Sun (Parker [8,9]). This indicates the rationale for some additional acceleration mechanism to operate beyond the coronal base. Parker [8] gave an ingenious stationary model which provided for the smooth acceleration of the solar wind through transonic speeds by continually converting the thermal energy into the kinetic energy of the wind. The solar wind was confirmed and its properties were recorded by in situ observations (Neugebauer and Snyder [10], Hundhausen [11], Meyer-Vernet [12]). The stellar rotation is found to lead to faster stellar winds and hence enable protostars and strong rotators to lose their angular momentum quickly via the mechanism of centrifugal and magnetic driving (Shivamoggi [13]).

One of the main assumptions in Parker’s [8] solar wind model is that the gas flow occurs under isothermal conditions (in standard notation),
(1)$$\begin{array}{cc} p & =a_{0}^{2}\rho \;, \end{array}$$
where $a_{0}$ is the constant speed of sound. However, the more realistic non-isothermal nature of the solar wind, thanks to a variable extended heating of the corona, may be represented in a first approximation by using the polytropic gas relation (Parker [14], Holzer [15], Keppens and Goedbloed [16]), (note that: Polytropic gas model provides a more general framework and is fully compatible with the adiabatic condition which is a special case of the polytropic gas model and corresponds to $\gamma =\frac{5}{3}$. The isothermal gas flow corresponds to γ=1),
(2)$$\begin{array}{cc} p & =C\rho^{\gamma }\;, \end{array}$$
where *C* is an arbitrary constant and γ is the polytropic exponent, 1<γ<5/3. In this paper, we give the modified Parker’s equation governing the acceleration of solar wind of a polytropic gas and rectify an apparent error in the formulation given previously by Holzer [15]. We then present a more compatible formulation and make detailed numerical and systematic asymptotic theoretical considerations to describe solar wind flow of a polytropic gas. The polytropic gas conditions are shown to lead to tenuous and faster solar wind flows and hence enable a quicker loss of the solar angular momentum.

## 2. Polytropic Gas Solar Wind Model

In Parker’s hydrodynamic model [8], the solar wind is represented by a steady and spherically symmetric flow, so the flow variables depend only on *r*, the distance from the Sun. (Note that: In reality, observations of the solar wind (Wang and Sheeley [17]) suggested and Kopp and Holzer [18] proposed a rapidly-diverging super-radial wind flow, especially in some active regions such as the coronal holes.) The flow velocity is further taken to be only in the radial direction—either inward (accretion model) or outward (wind model). We assume for analytical simplicity that the flow variables and their derivatives vary continuously so that there are no shocks anywhere in the region under consideration.

The mass conservation equation is
(3)$$\begin{array}{cc} \frac{2}{r}+\frac{1}{\rho }\frac{\;d\rho }{\;dr}+\frac{1}{v_{r}}\frac{\;dv_{r}}{\;dr} & =0\;. \end{array}$$

Assuming the gravitational field to be produced by the central solar mass $M_{s}$, Euler’s equation of momentum balance is
(4)$$\begin{array}{cc} \rho v_{r}\frac{\;dv_{r}}{\;dr} & =-\;\frac{\;dp}{\;dr}-\frac{GM_{s}}{r^{2}}\rho \;, \end{array}$$

*G* being the gravitational constant.

Using Equations (Equation 2) and (Equation 3), Equation (Equation 4) becomes
(5)$$\begin{array}{cc} \left(M^{2}-1\right)\frac{2}{v_{r}}\frac{\;dv_{r}}{\;dr} & =\frac{4}{r^{2}}\left(r-r_{*}\right)\;. \end{array}$$

In Equation (Equation 5), $r_{*}$ locates the Parker critical point and is given by
(6a)$$\begin{array}{cc} r_{*} & \equiv \frac{GM_{S}}{2a^{2}}\;, \end{array}$$
where a=a(r) is the variable sound speed in the polytropic gas described by Equation (Equation 2), and is given by
(6b)$$\begin{array}{cc} a^{2} & \equiv \frac{\;dp}{\;d\rho }=\frac{\gamma p}{\rho }\;, \end{array}$$
and *M* is the Mach number of the gas flow,
(6c)$$\begin{array}{cc} M & \equiv \frac{v_{r}}{a}\;. \end{array}$$

At a given point *r*, the local conservation of total energy (thermal plus kinetic energy) of a fluid particle gives,
(7)$$\begin{array}{cc} h+\frac{1}{2}v_{r}^{2}=\mathrm{const} & =h_{0}\;, \end{array}$$
where *h* is the enthalpy and the subscript 0 denotes the stagnation-flow values which are those that would result if the fluid particle were locally brought to rest isentropically and are assumed to be independent of *r*. This assumption becomes more and more sound as one goes past the Parker critical point and the effect of solar gravity on the solar wind becomes weak. Alternatively, $h_{0}$ may be taken to represent an average stagnation enthalpy over the coronal region of interest.

For a perfect gas, given by the equation of state (in standard notation),
(8)$$\begin{array}{cc} p & =\rho RT\;, \end{array}$$
we obtain on using Equation ([Disp-formula FD6b-entropy-23-01497]) (Shivamoggi [19])
(9)$$\begin{array}{cc} h & =\left(\frac{\gamma }{\gamma -1}\right)\frac{p}{\rho }=\frac{a^{2}}{\gamma -1}\;. \end{array}$$

Using Equation (Equation 9), Equation (Equation 7) becomes
(10)$$\begin{array}{cc} \frac{a^{2}}{\gamma -1}+\frac{1}{2}v_{r}^{2} & =\frac{a_{0}^{2}}{\gamma -1}\;, \end{array}$$
which leads to
(11)$$\begin{array}{cc} \frac{a^{2}}{a_{0}^{2}} & =\frac{1}{1+\left(\frac{\gamma -1}{2}\right)M^{2}}\;. \end{array}$$

As indicated by Equation (Equation 5), $r_{*}$ corresponds to the local sonic conditions (M=1), where
(12)$$\begin{array}{cc} a & =a_{*}\equiv a_{0}\sqrt{\frac{2}{\gamma +1}}\;, \end{array}$$

$a_{*}$, called the critical sound speed, is the local sound speed at the Parker critical point $r=r_{*}$. Conversely, the Parker critical point $r=r_{*}$ is determined by the condition $a=a_{*}$.

Using Equation (Equation 11), Equation (Equation 5) becomes
(13)$$\begin{array}{cc} \frac{\left(M^{2}-1\right)}{M^{2}\left[1+\left(\frac{\gamma -1}{2}\right)M^{2}\right]}\frac{\;d}{\;dr}\left(M^{2}\right) & =\frac{4}{r^{2}}\left(r-r_{*}\right)\;. \end{array}$$

Incidentally, it may be noted that by introducing the total energy
(14)$$\begin{array}{cc} E & \equiv \frac{v_{r}^{2}}{2}+\frac{a^{2}}{\gamma -1}-\frac{GM_{s}}{r}\;, \end{array}$$
the right-hand side of Equation (Equation 13) may be written alternatively as
(15)$$\begin{array}{cc} \frac{\left(M^{2}-1\right)}{M^{2}\left[1+\left(\frac{\gamma -1}{2}\right)M^{2}\right]}\frac{\;d}{\;dr}\left(M^{2}\right) & =\frac{4E+\left(\frac{4\gamma -6}{\gamma -1}-M^{2}\right)\frac{GM_{s}}{r}}{r\left(E+\frac{GM_{s}}{r}\right)}\;, \end{array}$$
which shows that the corresponding equation
(16)$$\begin{array}{cc} \frac{M^{2}-1}{M^{2}}\;\frac{\;d}{\;dr}\left(M^{2}\right) & =\frac{\left(1+\frac{\gamma -1}{2}M^{2}\right)\left(4E+\frac{3\gamma -5}{\gamma -1}\frac{GM_{s}}{r}\right)}{r\left(E+\frac{GM_{s}}{r}\right)}\;, \end{array}$$
given by Holzer [15] is apparently erroneous. (Note: As Holzer [15] noted, in the adiabatic gas flow ($\gamma =\frac{5}{3}$) case, Equation (Equation 16) does not allow smooth acceleration of the solar wind through transonic speeds. Equation (Equation 16) is also found to exhibit some anomalies in the solar wind velocity near the Sun ($\frac{r}{r_{*}}\ll 1)$). However, Equations (Equation 15) and (Equation 16) reduce to Parker’s [8] hydrodynamic equation in the isothermal-gas limit, γ→1.

Introducing
(17)$$\begin{array}{cc} r_{{*}_{0}} & \equiv \frac{GM_{s}}{2a_{0}^{2}}\;, \end{array}$$
and using Equation (Equation 11), Equation (Equation 13) may be rewritten as
(18)$$\begin{array}{cc} \frac{\left(M^{2}-1\right)}{M^{2}\left[1+\left(\frac{\gamma -1}{2}\right)M^{2}\right]}\frac{\;d}{\;dr}\left(M^{2}\right) & =\frac{4}{r^{2}}\left[r-\left\{1+\left(\frac{\gamma -1}{2}\right)M^{2}\right\}r_{{*}_{0}}\right]\;. \end{array}$$

The numerical solution of Equation (Equation 18) is plotted in Figure 1, which shows the polytropic gas conditions enhance the acceleration of the solar wind which is plausible because polytropic gas conditions favor an enhanced conversion of thermal energy in the solar wind into kinetic energy of the outward flow.

We now consider some special asymptotic cases for which Equation (Equation 18) facilitates simpler solutions.

### 2.1. Transonic Regime

Near the sonic critical point $r=r_{*}$, (note: In traditional gas dynamics, the transonic flow case is taken to mean M≈1.), we may write
(19)$$\begin{array}{cc} r & =r_{*}+x,\;M^{2}=1+y\;. \end{array}$$

Using Equations (Equation 11) and (Equation 17), Equation (Equation 18) then gives:
(20a)$$\begin{array}{cc} \left(y-\alpha x\right)\frac{\;dy}{\;dx} & =\beta x\;, \end{array}$$
where
$$\begin{array}{cc} \alpha & \equiv \frac{4}{r_{{*}_{0}}}\left(\gamma -1\right)\;,\\ \beta & \equiv \frac{4}{r_{{*}_{0}}^{2}}\;. \end{array}$$

Equation ([Disp-formula FD20a-entropy-23-01497]) is a first order ordinary differential equation of the *homogeneous* type, which may be recast as
(20b)$$\begin{array}{cc} \left(y-\alpha x\right)\;dy & =\beta x\;dx\;. \end{array}$$

We therefore introduce
(21)$$\begin{array}{cc} z & \equiv \frac{x}{y}\;. \end{array}$$

Equation ([Disp-formula FD20b-entropy-23-01497]) then becomes an ordinary differential equation of a *separable* type,
(22a)$$\begin{array}{cc} \left(1-\alpha z-\beta z^{2}\right)\;dy & =\beta yz\;dz\;, \end{array}$$
or
(22b)$$\begin{array}{cc} \frac{\;dy}{y} & =\frac{z}{\frac{1}{\beta }-\frac{\alpha }{\beta }z-z^{2}}\;dz\;. \end{array}$$

The solution of Equation ([Disp-formula FD22b-entropy-23-01497]) is:(23)$$\begin{array}{cc} y{\left[{\left(z+\frac{\alpha }{2\beta }\right)}^{2}-\left(\frac{1}{\beta }+\frac{\alpha^{2}}{4\beta^{2}}\right)\right]}^{\frac{1}{2}}{\left[\frac{\left(z+\frac{\alpha }{2\beta }\right)+\sqrt{\frac{1}{\beta }+\frac{\alpha^{2}}{4\beta^{2}}}}{\left(z+\frac{\alpha }{2\beta }\right)-\sqrt{\frac{1}{\beta }+\frac{\alpha^{2}}{4\beta^{2}}}}\right]}^{\frac{\frac{\alpha }{4\beta }}{\sqrt{\frac{1}{\beta }+\frac{\alpha^{2}}{4\beta^{2}}}}} & =0\;, \end{array}$$
from which,
(24a)$$\begin{array}{cc} y & =\left(\frac{1}{\sqrt{\frac{1}{\beta }+\frac{\alpha^{2}}{4\beta^{2}}}-\frac{\alpha }{2\beta }}\right)x\;, \end{array}$$
or
(24b)$$\begin{array}{c} \left(M^{2}-1\right)=\left(\frac{1}{\sqrt{\frac{1}{\beta }+\frac{\alpha^{2}}{4\beta^{2}}}-\frac{\alpha }{2\beta }}\right)\left(r-r_{*}\right)\;. \end{array}$$

Equation ([Disp-formula FD24b-entropy-23-01497]), numerically displayed in Figure 2, shows an enhanced acceleration of the polytropic wind (γ>1) past the Parker critical point, in agreement with the numerical solution of Equation (Equation 18). (Note: In the near isothermal-gas case, γ≈1, putting ε=γ−1>0, Equation ([Disp-formula FD24a-entropy-23-01497]) becomes $y\approx 2\left(1+\varepsilon \right)\frac{x}{r_{{*}_{0}}}$ which indicates again an enhanced acceleration of the polytropic gas flow past the Parker critical point).

### 2.2. Near-Sun Regime

For $r/r_{*}<<1$, Equation (Equation 18) may be approximated by
(25)$$\begin{array}{cc} \frac{1}{M^{2}{\left[1+\left(\frac{\gamma -1}{2}\right)M^{2}\right]}^{2}}\frac{\;d}{\;dr}\left(M^{2}\right) & \approx \frac{4r_{{*}_{0}}}{r^{2}}\;, \end{array}$$
from which
(26)$$\begin{array}{cc} \frac{M^{2}}{1+\left(\frac{\gamma -1}{2}\right)M^{2}} & \approx e^{-\left(4r_{{*}_{0}}/r\right)}\;. \end{array}$$

In the isothermal limit (γ→1), Equation (Equation 26) leads to
(27)$$\begin{array}{cc} M^{2} & \approx e^{-\left(4r_{{*}_{0}}/r\right)}\;. \end{array}$$

Comparison of Equation (Equation 26) with Equation (Equation 27) shows again an enhanced acceleration of the polytropic solar wind (γ>1) near the Sun.

### 2.3. Far-Sun Regime

For $r/r_{*}>>1$, Equation (Equation 18) may be approximated by
(28)$$\begin{array}{cc} \frac{\;d}{\;dr}\left(M^{2}\right) & \approx \frac{2\left(\gamma -1\right)}{r}M^{2}\;, \end{array}$$
from which follows that
(29)$$\begin{array}{cc} M^{2} & \approx r^{\left[2\left(\gamma -1\right)\right]}\;. \end{array}$$

Equation (Equation 28) shows again an enhanced acceleration of the polytropic solar wind (γ>1) far from the Sun.

### 2.4. Effective de Laval Nozzle for a Polytropic Solar Wind

The continuous acceleration of the polytropic solar wind flow, seen above, from subsonic speeds at the coronal base to supersonic speeds away from the Sun implies a *de Laval* nozzle-type situation operational near the Sun (Clauser [20]).

If A=A(r) is the cross section area of an effective de Laval nozzle associated with the polytropic solar wind flow, we have from equations Equations (Equation 5) and (Equation 13)
(30)$$\begin{array}{cc} \frac{\left(M^{2}-1\right)}{M^{2}}\left(\frac{2v_{r}}{a^{2}}\right)\frac{\;dv_{r}}{\;dr} & =\frac{\left(M^{2}-1\right)}{M^{2}\left[1+\left(\frac{\gamma -1}{2}\right)M^{2}\right]}\frac{\;d}{\;dr}\left(M^{2}\right)\\ \; & =\frac{2}{A}\frac{\;dA}{\;dr}\\ \; & =\frac{4}{r^{2}}\left(r-r_{*}\right)\;, \end{array}$$
from which follows that
(31)$$\begin{array}{cc} A(r) & =4\pi r^{2}e^{-2r_{{*}_{0}}\int_{r_{o}}^{r}\frac{1}{r^{2}}\left[1+\left(\frac{\gamma -1}{2}\right)M^{2}\right]\;dr}\;, \end{array}$$
where $r=r_{0}$ at the coronal base. Equation (Equation 31) implies that the cross section area of the effective de Laval nozzle for the polytropic solar wind (γ>1) varies faster, in consistency with the enhanced acceleration of the polytropic solar wind.

## 3. Discussion

Thanks to the more realistic non-isothermal nature of the solar wind that can be caused by a variable extended heating of the corona, a polytropic gas model is in order to describe the solar wind flow. In recognition of this, the present paper is aimed at putting forward a compatible theoretical formulation and providing detailed numerical and systematic asymptotic theoretical considerations to describe acceleration of the solar wind of a polytropic gas. The polytropic gas conditions have been shown to lead to tenuous and faster solar wind flows because polytropic gas conditions favor an enhanced conversion of thermal energy in the solar wind into kinetic energy of the outward flow, hence enabling a quicker loss of the solar angular momentum.

On the other hand, it may be mentioned that explicit investigation of the effects of coronal heat addition has been made numerically (Leer et al. [21,22]). Though some coronal heating scenarios (caused by a variable thermal conductivity or dissipation of waves) may rather qualitatively simulate polytropic gas effects in the solar wind, a coronal heat addition is not explicitly included in our present formulation. So, it is not possible to identify clear relationships between the parameters characterizing the two scenarios and, hence, to make definitive comparisons of our present theoretical results on the polytropic gas effects with the numerical results on the effects of explicit inclusion of coronal heat addition.

## Figures and Tables

**Figure 1 entropy-23-01497-f001:**
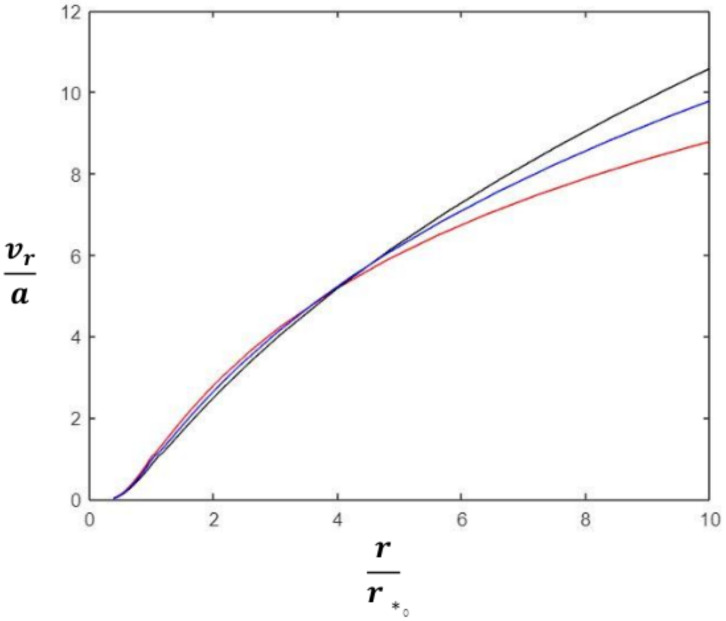
Effect of re-normalized polytropic exponent on solar wind speed. (i) Parker model $\hat{\alpha }=0$ (red), (ii) polytropic gas model $\hat{\alpha }=0.05$ (blue), (iii) polytropic gas model $\hat{\alpha }=0.1$ (black). Here, $\hat{\alpha }\equiv \frac{\gamma -1}{2}$.

**Figure 2 entropy-23-01497-f002:**
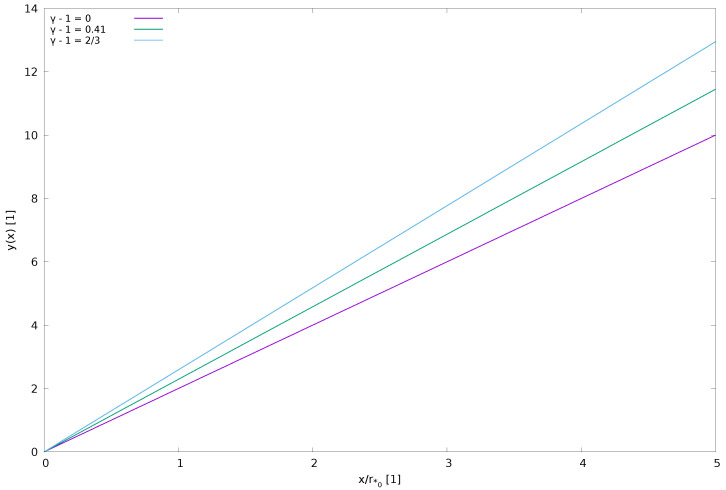
Acceleration of polytropic wind (Equation (Equation 17)) for various values of the polytropic exponent (two limit values and one for Hydrogen).

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
