# Peer review of "Parker’s Solar Wind Model for a Polytropic Gas"

_entropy, 2021, doi:10.3390/e23111497_

Round 1

Reviewer 1 Report

Comments on the paper “Parker’s Solar Wind Model for a Polytropic Gas” by Bhimsen Shivamoggi, David Rollins and Leos Pohl.

This paper deals with the MHD equilibrium equations that describe the solar wind acceleration in proximity of solar corona. In particular, this paper presents a generalization of Parker’s 1958 hydrodynamic solar wind model, which consists of stationary 1–dimensional (spherically symmetric) equilibrium equations, under isothermal conditions. Here, instead of assuming the plasma to be isotropic as in Parker’s model, a polytropic  gas equilibrium equation is adopted. This new formulation with  polytropic relation results in enhanced acceleration of the solar wind, reaching the sonic critical condition lower in the corona. Also the supersonic acceleration is stronger, suggesting a faster loss of mass and angular momentum for the sun.

The paper is well written, all the equations introduced well explained, and the reasoning is clear. I suggest publication as it is.

Author Response

Since the reviewer suggests publication of the paper as "was", we have no response to the reviewer's notes.

Reviewer 2 Report

The manuscript is concerned with the problem determining physical parameters in solar wind during its expansion in the heliosphere, in particular, the expansion velocity. The authors use a hydrodynamic model where a polytropic law is assumed. Relevant equations are derived that are numerically solved determining the radial profile of the wind velocity for various values of the polytropic index gamma. Approximate solutions in limited regions (close and far to the Sun, and across the sonic point) are analytically calculated. The manuscript gives a valuable contribution to the problem of describing the acceleration of solar and stellar winds and of angular momentum loss in astrophysical objects. In particular, its main purpose appears to be rectifying previous results found by Holzer (ref. [15]). On the whole, the manuscript is clearly written, even if some points need to be clarified. Therefore, I believe that I could be published as soon as the following points will be addressed by the authors:

1) Concerning the definitions given in Eq.s (6): in the isothermal case (gamma=1) the quantity "a" (representing the sound speed) is a constant. In contrast, in the polytropic case "a" depends on rho (or on p). Since rho changes with the radial distance r, then "a" is a function of r, and r_* is a function of r, too. In this respect r_* does not represent the critical radius (which should be a constant independent of r). This fact should be explicitly emphasized to avoid confusion.

2) The quantity a_0 in eq. (7) is defined in the note 5 as a quantity corresponding to the "stagnation flow condition". However, since results that follows depend on a_0 (e.g., the quantity r_{*0}=G M_s/2 a_0^2),  it would be desirable to have an explicit expression for a_0. In particular, is a_0 a constant quantity (independent of r)? If a_0 is constant, then r_{*0} is constant, too. Does r_{*0} represent the critical radius?

3) In the note 5 it is stated that Eq. (7) is universally valid and the reader is lead to the reference [19] for a derivation. However, more details in the manuscript about how Eq. (7) is derived would help the reader to understand the whole development. By the way, ref. [19] is a book, which could be difficult to be accessed by readers.

4) Fig. 1: the label indicating the quantity on the horizontal axis is r/r_*. However, r_* is in general a function of r, except in the isothermal case (see point 1 above). I guess that the quantity on the horizontal axis is r/r_{*0}. Please, clarify.

5) The cases gamma=1 and gamma=5/3 corresponds to the isothermal and adiabatic case, respectively. Therefore, decreasing gamma from 5/3 to 1 corresponds to increase heat deposition in the wind (e.g., due to turbulent dissipation). Therefore, results of Fig.s 1 and 2 show that increasing heat deposition would decrease the wind acceleration, at least in the supersonic region. Moreover, in the Conclusions it is stated that "for the polytropic case (gamma > 1), the sonic critical point occurs lower in the corona"; in other words, a lower heat deposition corresponds to a critical point closer to the Sun.
However, these results seems to be in contrast with what found by Lee and Holzer, JGR, 85, 4681 (1980). Those authors discuss a case when heat is added to the plasma in the supersonic region, finding that a larger heat deposition gives a larger flow speed at 1 AU (see also Leer et al., Space Sci. Rev., 33, 161 (1982), page 171), and moves the sonic point inward.
Are the rusults by Lee and Holzer (1980) biased by the same error found in ref. [15]? Can the authors discuss such a point?

6) Even if the manuscript is mainly devoted to rectify results found by Holzer [15], a wider discussion on how the present results compare with the literature on the same subject would be desirable.

Minor points:

a) The quantity alpha defined just after Eq. (15a) has the same meaning of alpha=(gamma-1)/2 indicated in the caption of fig. 1? If not, please use different symbols for the two quantities. 

b) Fig. 2: numbers and letters on the axes are too small. Please, magnify them.

Round 2

Reviewer 2 Report

In the new version of the manuscript the authors have taken into account all the suggestions I gave in my previous report, clarifying all the points. The theoretical development now appears to be much clearer. Therefore, in my opinion the manuscript is now in a suitable form to be published.